# First Records of Heartbeats via ECG in a Stingless Bee, *Melipona flavolineata* (Apidae, Meliponini), during Contention Stress Using Isoflurane as an Anesthetic

**DOI:** 10.3390/insects14080696

**Published:** 2023-08-08

**Authors:** Felipe Andrés León Contrera, Bárbara dos Santos Conceição Lopes, Clarissa Araújo da Paz, Maria Klara Otake Hamoy, Murilo Farias dos Santos, Gabriela Brito Barbosa, Anthony Lucas Gurgel do Amaral, Luiz Henrique Barbosa de Pinho, Moisés Hamoy

**Affiliations:** 1Laboratório de Biologia e Ecologia de Abelhas, Instituto de Ciências Biológicas, Universidade Federal do Pará, Belém 66075-110, PA, Brazil; barbara91.abelha@gmail.com; 2Laboratório de Farmacologia e Toxicologia de Produtos Naturais, Instituto de Ciências Biológicas, Universidade Federal do Pará, Belém 66075-110, PA, Brazil; clarissa.paz@icb.ufpa.br (C.A.d.P.); klara04hamoy@gmail.com (M.K.O.H.); murilofarias329@gmail.com (M.F.d.S.); gabriela.bribarbosa@gmail.com (G.B.B.); anthony.amaral@ics.ufpa.br (A.L.G.d.A.); pirespique1@gmail.com (L.H.B.d.P.)

**Keywords:** electrophysiology, bee, *Melipona*, ECG, restraint stress, anesthetic

## Abstract

**Simple Summary:**

Changes in the heart activity of bees have been correlated with several stressors. However, this has been studied in just a few species within Apoidea. The purpose of our study was to evaluate several parameters of heart activity in a stingless bee species from the Amazon region, *Melipona flavolineata,* during a 30 min stress period. Workers of the species were restrained and anesthetized using isoflurane. The basal heartbeat at the first five minutes of the records was about 600 beats per minute, which generally increased over the restraining period. The contention stress also changed other parameters of heart activity in the species. Our study was the first to evaluate the cardiac activity in a stingless bee, an important group of tropical pollinators, and showed that heart activity was altered by stress. Thus, heart monitoring using ECG is a feasible technique to show some of the effects of different stressors, such as pesticides, in the physiology of worker bees. We also showed that isoflurane can be used as an anesthetic, an alternative to narcosis induced by CO_2_ and cold, although its effects must still be studied.

**Abstract:**

The hemodynamic activity of *Melipona flavolineata* workers was evaluated during restraint stress for a period of 30 min. The observed parameters were power variation in the elapsed time, and subsequently, six periods of one second were divided and called A, B, C, D, E and F; in each period, the electrocardiographic parameters were evaluated: spike frequency, amplitude, spike intervals and spike duration. The experiment was carried out with eight worker bees of *M. flavolineata*, for which electrodes of a nickel–chromium alloy were made. The bees were previously anesthetized with isoflurane and properly contained and fixed in a base for stereotaxis in which the electrode was implanted. All these procedures were performed inside a Faraday cage. The results showed power oscillations during the recording, with the highest energy level being between 300 and 600 s. Spike frequency, spike amplitude, interval between spikes and spike duration parameters underwent changes during the restraint stress period. Thus, the cardiac activity of *M. flavolineata* can be used as a biomarker and can be used to clarify physiological issues or alterations caused by toxic agents and indicate risk factors for these animals.

## 1. Introduction

The utilization of the electrophysiological method is an important method for improving the understanding of the mechanisms of ion displacement to the epithelial tissues. These mechanisms are analyzed as electronic circuits and promoted through two electrodes that are placed close to the heart and displayed as a graph or waveform to detect and record the electrical signals produced during each cardiac cycle [1,2]. The electrocardiogram (ECG) monitors the heartbeat rhythm and cardiac frequency and identifies heart diseases through electrophysiological analysis [3].

Arthropoda (Ecdysozoa, Arthropoda) possesses an open circulatory system, divided into two parts: the vascular and the lacunar. The vascular portion is located in the medial line of the arthropods and consists of a tubular heart as a central element, which is located right below the dorsal α-chitin cuticle and bombs hemolymph to their different body parts. The dorsal vessel is myogenic, but its rhythmicity is modulated by neuropeptides and neurotransmitters [4]. The venous return occurs in the lacunar portion, where the hemolymph flows directly from the hemocoel to the vascular portion, passing by the ostia [5]. Hemolymph circulation is crucial for promoting the transport of nutrients, neurohormones, molecules of circulation, immunological factors and residual products [6].

In bees (Hymenoptera, Apoidea), a taxon that provides important ecosystem services, mainly pollination [7,8], the vascular portion consists of the heart (located in the abdomen) and the aorta (located in the thorax), the diameter of which is much smaller [9,10]. The bee heart is tubular, located below the abdominal wall; it begins in the first abdominal ostia and extends up to the posterior half of the VI abdominal segment and possesses several openings (ostioles) in the wall, which allows the entrance of hemolymph from the lacunar portion. In the anterior part, it connects with the aorta; the hemolymph flows from the final part of the heart, in the abdomen, in the direction of the thorax. The heart has a muscular wall that pumps the hemolymph to the aorta, which is a thinner vessel that pumps the hemolymph to the heart; in this area, it possesses an opening in its extremity, below the heart [10,11]. The bee circulatory system has contractions both in the dorsal vessel and in pulsating organs that provide flow for the extremities [12]. These contractions are caused by diaphragms, which causes the hemolymph to return from the hemocoel to the heart, via the ostioles [10].

Studies about the cardiac function and blood (hemolymph) circulation in bees are relatively scarce and have been mostly performed in the honeybee, *Apis mellifera*, and to a lesser extent, in the bumblebee, *Bombus terrestris*. These studies are mostly descriptive, while more recent studies have focused on the effects of stressors, mainly pesticides, on the heart and other functions [12,13,14,15,16,17,18,19,20]. Among these studies, few used ECG for measuring the bees’ heartbeats [18,19].

In stingless bees [21], a taxon of eusocial bees consisting of more than 600 species [22], there are no studies on their cardiac function. Stingless bees are important pollinators [7] that are exposed to several stressors, such as thermal stress caused by global climatic changes [23], by natural foraging behavior [24] and by pesticides [25]. A recent study showed that the effects of pesticides, for example, dimethoate, do not equally affect stingless bees and *A. mellifera* [26]; however, see [27] for a rebuttal. Thus, base studies on Meliponini, including their cardiac function, are important for the understanding of their physiology and how they could respond to different stressors. For example, in *A. mellifera*, it was shown that amitraz, an acaricide used to control the mite *Varroa destructor*, provokes an acceleration of the heartbeats of workers compared with the control treatment [18].

Thus, in this study, we studied the cardiac function, via ECG, of the Amazonian stingless bee species *Melipona flavolineata* Friese, 1900 [28], an important pollinator and visitor of native and crop plants [7,29] and used as a model species for studies on the reproduction and supplemental feeding of stingless bee colonies [30,31,32]. We evaluated the heartbeat in the studied species and the effect of contention stress in bees across time. We measured the heartbeats for 30 min, observing the amplitude, spike frequency, energy intensity, spike intervals and the spike durations, during the restraint period. We used isoflurane as an anesthetic, which has been used successfully in insects [33,34] and bees, causing few long-term alterations in their physiology when compared to other methods of sedation [34,35].

## 2. Materials and Methods

### 2.1. Studied Species

We used eight *M. flavolineata* worker bees, a species that inhabits the Brazilian states of Pará, Maranhão and Tocantins [28], whose colonies were maintained in the meliponary of the Laboratory of Biology and Ecology of Bees of the Federal University of Pará, Belém, Brazil. We chose young worker bees from the nest that were not still engaged in forager activity. These bees were brought to the Laboratory of Pharmacology and Toxicology of Natural Products (Laboratório de Farmacologia e Toxicologia de Produtos Naturais) UFPA-ICB and kept in an environment with a regulated temperature (25–28 °C) and were used in the experiments described below.

### 2.2. Animal Preparation for the Experiment

Bees were previously anesthetized with isoflurane at a room temperature of 24 °C, with a 1.5 mL volume of the drug soaked in cotton, and they were kept in contact with the anesthetic within a 150 × 20 mm Petri dish glass until the loss of the postural reflex onwards (Figure 1A). The mean time for induction was 57.75 ± 8.44 s, and the mean time for recovery of the posture reflex was 2645 ± 33.56 s (Figure 1B). This procedure was necessary to allow the fixation of the bees upon a polyethylene foam platform with a small rubber band between the bees’ thorax region and the abdomen. Thus, it was possible to fix the electrodes for the ECG (Figure 1C). Pins prevented the bee from moving too much and interfering with the measurements. In this work, the method by Kaiser [36] was used, with modifications, to capture the data with the available equipment and materials.

### 2.3. Manufacture of Electrodes, Implantation and Obtaining of Electrocardiographic (ECG) Recordings

Regarding the ECG, the electrodes were made from JST SM cables with 2.13 cm long Jack pins. The nickel–chromium wire electrodes (Morelli ortodontia), designed solely for measuring the heartbeats in the studied species, were conjugated at a distance of 1 mm, with 0.2 mm in diameter and 2 mm in length. They were insulated with a liquid insulator, and after drying the material, the electrode was fixed in a stereotactic device. After the bee’s fixation, the following coordinates were obeyed, considering the recording electrode as a parameter: the zero point was at the intersection between thorax and abdomen in the midsagittal line, with an anteroposterior coordinate of 1 mm, and punched the abdomen, thus touching the dorsal vessel at the dorsoventral coordinate of 0.6 mm of depth (Figure 2). After the procedure, the ECG was recorded. The entire procedure for obtaining the record was performed inside a metal screen Faraday cage. The electrodes were connected to a high-impedance amplifier (Grass Technologies, 600 East Greenwich Avenue, West Warwick, RI 02893 USA) with a signal amplification of 50,000×, monitored by an oscilloscope (Protek, 6510, Gwangmyeong-si, Republic of Korea). Data were stored and digitized on a National Instruments board.

All recordings lasted 30 min for each bee. Three fragments, lasting 1 s, of the records were taken every 5 min for analysis and comparison of the effects of stress on hemolymph pumping. Thus, we analyzed the following periods: A = 99–100, 199–200 and 299–300 s; B = 399–400, 499–500 and 599–600 s; C = 699–700, 799–800 and 899–900 s; D = 999–1000, 1099–1100 and 1199–1200 s; E = 1299–1300, 1399–1400 and 1499–1500 s; and F = 1599–1600, 1699–1700 and 1799–1800 s. This delimitation of periods was necessary due to the large database generated, which would be difficult to visualize if shown continuously. Data were continuously stored at 1 KHz in 3 KHz low pass and 0.3 Hz high pass, and the signal acquisition process was performed automatically. Analyses were performed using commercial Labview software (Part No. 779448–35).

### 2.4. Statistical Analyses

For statistical analysis, normality and homogeneity tests were performed for data variations using the Kolmogorov–Smirnov and Levene tests, respectively. Data are presented as mean ± standard deviation (SD), and *F* and *p* values are included, when relevant. Significance levels of * *p* < 0.05, ** *p* < 0.01, *** *p* < 0.001 were considered for all analyses. Since the results between the animals were very similar in the morphographic elements, the comparisons between analyzed periods were based on a two-way ANOVA, followed by Tukey’s test for multiple comparisons. The parameters analyzed in the records were spike frequency (Spikes min^−1^), amplitude of record (mV), spike intervals (ms) and spike durations (ms) (Figure 3B). Statistical analyses to identify and remove outliers and to evaluate the mean, spike intervals (ms), spike durations (ms) and frequency were performed using GraphPad Prism, version 8 (Graph-Pad Software Inc., San Diego, CA, USA).

## 3. Results

We observed that the triggering of impulses maintains the rhythm and can undergo rapid variations. When analyzing the records, we identified only spikes with an ascending phase (depolarization) and a descending phase (repolarization), which represents the contraction and relaxation of the structure (Figure 3A,B).

The electrocardiographic recording demonstrating the cardiac function of *M. flavolineata* (n = 8) in the first five minutes of recording during restraint showed a mean spike frequency of 605.9 ± 72.11 Spikes min^−1^, with an amplitude of 1.872 ± 0.829 mV (magnification of 50,000×). Spike intervals, which represent the positive trigger in the contraction, lasted 101.0 ± 11.01 ms. The duration of the spike, which indicates the time required to maintain hemolymph transport during contraction, averaged 6.421 ± 0.45 ms.

The ECG recordings lasting 30 min during *M. flavolineata* containment demonstrated amplitude variation (Figure 4A). Fragments were taken from each record for analysis at the end of every 5 min of recording to compare the effects of stress on hemolymph pumping. During the first five minutes, there was an increase in the amplitude of the recording (A), followed by a progressive decrease until the 10th minute of recording (B). The amplitude increased again at the 15th minute (C) and showed little variation in the fragments of the 20th (D), 25th (E) and 30th (F) minutes. The spectrogram demonstrates an increase in the level of circulating energy in the period from 300 to 600 s (Figure 4B).

During the 30 min cardiac recording, a difference in the average linear power of the records could be observed, with an average power of 6.606 ± 0.746 mV^2^/Hz × 10^−3^ being obtained in the recording period of 0–300 s, which was significantly lower than in the periods of 300–600 s (33.15 ± 3.37 mV^2^/Hz × 10^−3^), 600–900 s (15.03 ± 1.64 mV^2^/Hz × 10^−3^) and 900–1200 s (12.36 ± 2.12 mV^2^/Hz × 10^−3^). However, it did not present a significant difference between the periods of 1200–1500 (5.602 ± 0.921 mV^2^/Hz × 10^−3^; *p* = 0.9018) and 1500–1800 s (4.600 ± 1.571 mV^2^/Hz × 10^−3^; *p* = 0.3179). The period of greatest power was 300–600 s, which was significantly higher than the other groups. In the periods of 600–900 s and 900–1200 s, the power remained constant and later showed a decrease in the periods of 1200–1500 s and 1500–1800 s (Figure 4C).

For each alteration in the tracing during the containment stress, a behavioral pattern of cardiac activity was analyzed (Figure 5). Thus, the spike frequency in period A (mean of 605.9 ± 72.11 Spikes min^−1^) differed from that in periods B (761.5 ± 97.45 Spikes min^−1^), C (737.4 ± 121.6 Spikes min^−1^) and D (762.0 ± 86.83 Spikes min^−1^); moreover, there was no difference for E (623.7 ± 76.82 Spikes min^−1^, *p* = 0.984), and it was greater than F (527.0 ± 80.56 Spikes min^−1^). Periods B, C and D showed a difference from groups E and F. Groups E and F showed a difference in the frequency of spikes (Figure 6A).

The amplitude of the spikes varied during the retention stress period. In periods A and B, there was an increase in amplitude ((1.872 ± 0.829 mV) and (1.722 ± 0.764 mV) (*p* = 0.886)), which was significantly higher than the other periods: C (0.941 ± 0.188 mV), D (0.803 ± 0.203 mV), E (0.809 ± 0.109 mV) and F (0.874 ± 0.126 mV) (Figure 6B).

For the spike intervals in period A (mean of 101.0 ± 11.01 ms), there was no significant difference for periods B (87.92 ± 7.168 ms; *p* = 0.069) and C (104.3 ± 14.58 ms; *p* = 0.089). However, they were higher than in period D (84.79 ± 7.501 ms). Period A did not differ from period E (110.5 ± 26.42 ms; *p* = 0.338). However, it was smaller than period F (114.9 ± 21.71 ms). Periods E and F showed statistical difference for periods B and D (Figure 6C). During the containment stress of worker bees, there was an increase in the duration of the spike from period D (mean: 11.41 ± 1.35 ms), E (10.52 ± 1.623 ms; *p* = 0.0789) and F (12.69 ± 3.268 ms) which was significantly higher than in periods A (6.42 ± 0.455 ms), B (6.763 ± 1.244 ms) and C (6.763 ± 1.244 ms). Period E was better than period F. Periods A, B and C did not show statistical differences (*p* = 0.989) (Figure 6D).

## 4. Discussion

Our study was the first to register in detail the cardiac function, via ECG, in a meliponine bee and to use isoflurane as an anesthetic, instead of CO_2_ or cold exposure. We showed in this study that the bee restraint caused variations in the heart activity of worker bees, with the highest potencies observed between 300 and 600 s; this suggests that more effort was required to pump the hemolymph in this period. The alterations in the bees’ heartbeats were higher according to the contention time; the other parameters of cardiac activity also changed in a similar way, mainly the amplitude, which varied during the 30 min recording. The experiment allowed us to evaluate the cardiac activity at rest, which is around 605 Spikes min^−1^, together with other factors.

The stress caused by a period of 30 min of restraint allowed us to verify alterations in *M. flavolineata* cardiac activity; the main functions that pointed to stress were the spike frequency, the amplitude and the parameter of contractile activity, which is related to the spike duration (the speed of heart contraction).

In *A. mellifera*, in vivo experiments showed that cold-anesthetized workers exposed to amitraz, an acaricide used to control infestations of *Varroa destructor* (Arachnida: Parasitiformes), had a significant increase (about 135%) in their heartbeat over several hours, shown by ECG [18]. A similar pattern was also found in another study, which also showed that, due to the interaction between amitraz and the octopamine receptors, there is a reduction in the tolerance of newly emerged workers to viral infections [16]. In this case, the intrinsic activities of the toxic element (amitraz) do not allow an evaluation of the cardiac parameters caused by the manipulation of bees. In our study, the contention stress allowed us to show the cardiac activity of the workers without the interference of any toxic agent. Thus, these studies showed that the cardiac function is an indicative parameter of physiological stress in bees [16,18]. Other stressors, such as pesticides, provoke noxious effects in stingless bees; however, cardiac function has not been yet included as a measured parameter in this group [37].

Another approached aspect of our study, also not tested in stingless bees, was the anesthetic effect of isoflurane. It has been previously tested in other bee species, presenting few lasting effects on their physiology [34,35]. In *M. flavolineata*, after one minute of anesthesia induction, workers returned to motor activity in about four to five minutes; after this period, they resumed motor activity (visualized by the head and antennal movement). Only after this period did we measure their cardiac activity. Our results are promising, since they showed that workers anesthetized with isoflurane “awoke” rapidly after the procedure and moved similarly to non-anesthetized bees. Since compelling evidence from various studies has shown the adverse short- and long-term effects of alternative anesthetics, such as CO_2_-induced and cold-induced narcosis, on *A. mellifera* workers [34,38,39,40,41], it is important that future research compare the physiological impacts of diverse anesthetics, including isoflurane, on stingless bees in an effort to gain a deeper understanding of the potential consequences that different anesthetic agents may have on the intricate physiology of stingless bees. This knowledge is crucial to developing more informed and judicious approaches to anesthesia in order to safeguard the well-being and long-term health of these important pollinators.

Moreover, it is essential to explore alternative approaches of restraint in future studies to thoroughly examine the impact of stress on the cardiac function of healthy worker bees. This is because the current method, which involves affixing the bees to a polyethylene foam platform using a rubber band placed between the thorax region and the abdomen, may potentially affect the heart function of the insects in comparison to unrestricted bees. Thus, it is crucial to investigate additional restraint techniques to gain a comprehensive understanding of the relationship between stress and cardiac performance in bees.

Therefore, our study has established a fundamental framework that enables the utilization of cardiac function as a reliable proxy variable to assess the impact of various stressors on stingless bees. By gaining a deeper understanding of the physiological cardiac function in arthropods, particularly in species like *M. flavolineata* and other bee species, through comprehensive electrocardiogram analyses, we can effectively investigate the effects of pesticides and other artificial substances on pollinators. This knowledge is crucial, as it directly relates to the well-being of these insects and their pivotal role in both human crop production and the reproductive mechanisms of wild flora. Consequently, our research contributes significantly to the ongoing discourse surrounding the health of pollinators and their profound influence on the sustainability of ecosystems.

Data on the cardiac function in other Meliponini species must also be obtained for comparisons, such as the dynamic of the hemolymph flow. The phenomenon of periodic reversal of the heartbeat, accompanied by the subsequent reversal of hemolymph flow, has been documented in various insect species across different stages of their life cycle [42,43,44,45]. However, in ancestral winged insects (pterygotes), hemolymph flow is only unidirectional, and in more recent (derived) groups, the pumping might be anterograde (from the abdomen to the head) or retrograde (from the head to the abdomen) [42,44]. The available information in bees is restricted to some species of *Bombus* spp. and *A. mellifera* [10] and shows that the pumping is anterograde. However, there is no available information for other bee species, or whether there a reversion of heartbeat might occur, as it occurs in some *Drosophila* species [43,45]. Given that hemolymph flow in *Bombus* species has been found to be intricately linked to individual thermoregulation prior to flight [44], it becomes crucial to gain a comprehensive understanding of blood circulation mechanisms in order to elucidate the intricate processes governing homeostasis in bees. This understanding includes investigating the neural components involved in regulating heart rate and directing insect cardiac function, as well as exploring the influence of diverse ion channels, cardiomodulatory peptides, neurotransmitters and whether the origin of this insect’s heartbeat is myogenic, originating from within the heart muscle itself, or neurogenic, originating from neural control over insect cardiac function and heart rate modulation. By delving into these scientific inquiries, we can unlock valuable insights into the interplay between physiological processes in bees, advancing in our understanding of their cardiovascular systems and the maintenance of vital homeostasis.

## Figures and Tables

**Figure 1 insects-14-00696-f001:**
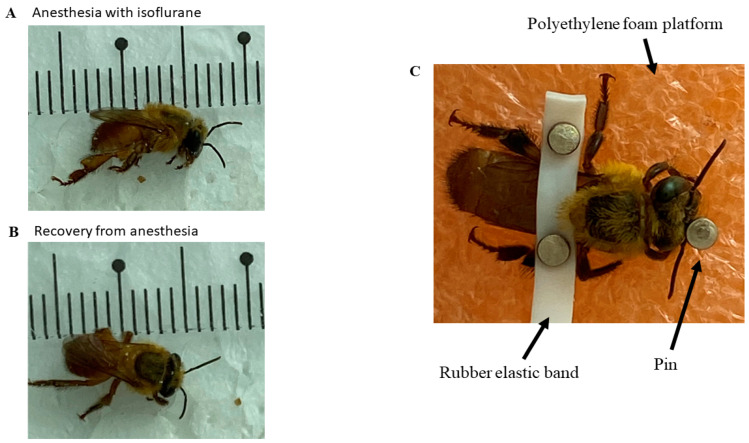
(**A**) Characteristics of *Melipona flavolineata* worker anesthetized with isoflurane and with loss of posture reflex; (**B**) recovery of posture reflex after anesthesia; (**C**) animal restrained for implantation of electrodes by stereotaxis.

**Figure 2 insects-14-00696-f002:**
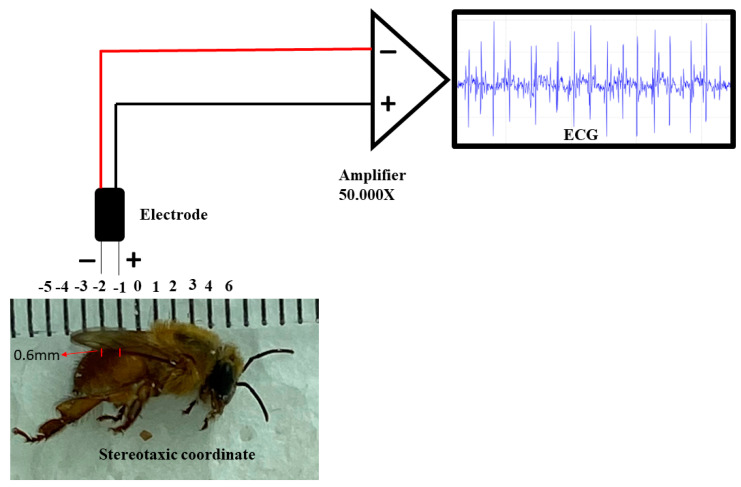
Components for the recording: stereotaxic coordinates used in the animals for the acquisition of the electrocardiographic record, positioning of the conjugated electrodes at a distance of 1 mm, high impedance amplifier (50,000× of signal amplification). The characteristic signal was observed on a monitor.

**Figure 3 insects-14-00696-f003:**
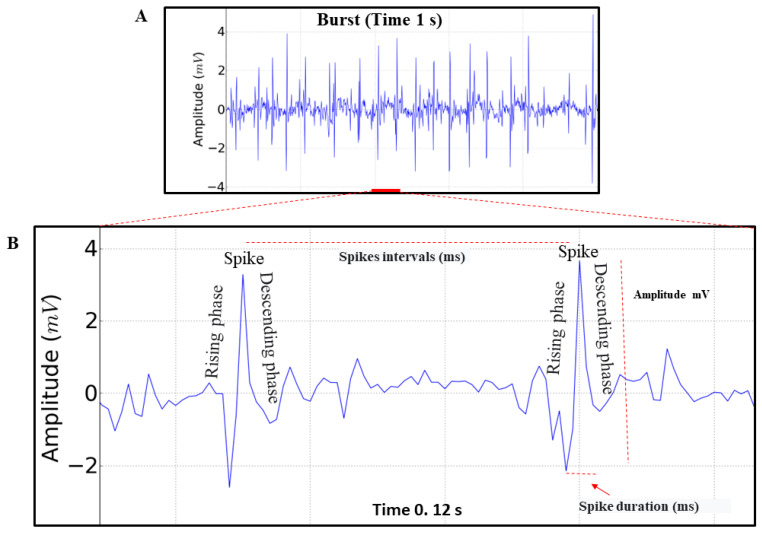
(**A**) Electrocardiogram of *Melipona flavolineata*, represented by 1 s of recording (potential burst), in the first 5 min of restraint; (**B**) amplification of 0.12 s of the ECG trace demonstrating the analyzed parameters that allow the evaluation of spike frequency (Spikes min^−1^), amplitude of record (mV), spike intervals (ms) and spike durations (ms).

**Figure 4 insects-14-00696-f004:**
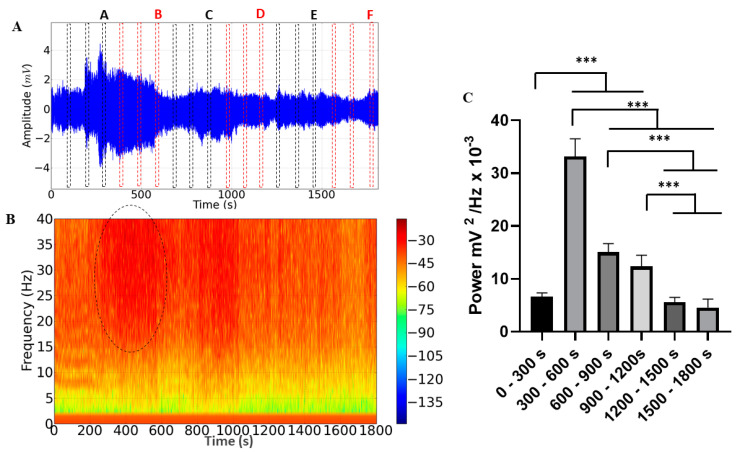
Electrocardiographic tracing represented by 30 min of recording, demonstrating the areas of the recording that were analyzed (dotted in red and black): A, B, C, D, E, and F. (**A**) Energy distribution spectrogram demonstrating areas with different cardiac energy intensities during restraint stress. The area demarcated by the dotted circle indicates the highest energy intensity of the record. (**B**) Linear power plot of cardiac activity during restraint stress in *Melipona flavolineata* (**C**) (after ANOVA followed by Tukey; *** *p* < 0.001; n = 8).

**Figure 5 insects-14-00696-f005:**
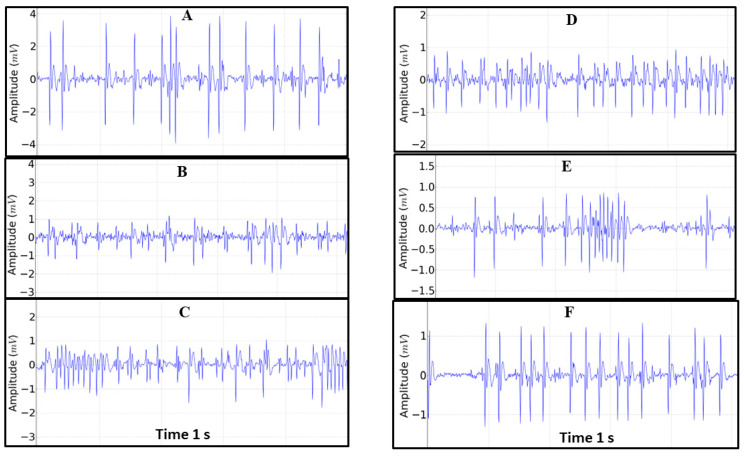
Tracing patterns found during the 30 min recording obtained every 5 min referring to the following patterns: (**A**) = 299–300 s, (**B**) = 599–600 s, (**C**) = 899–900 s, (**D**) = 1199–1200 s, (**E**) = 1499–1500 s and (**F**) = 1799–1800 s. In each tracing, the spike frequency (Spikes min^−1^), amplitude (mV), spike intervals and spike durations (ms) were analyzed.

**Figure 6 insects-14-00696-f006:**
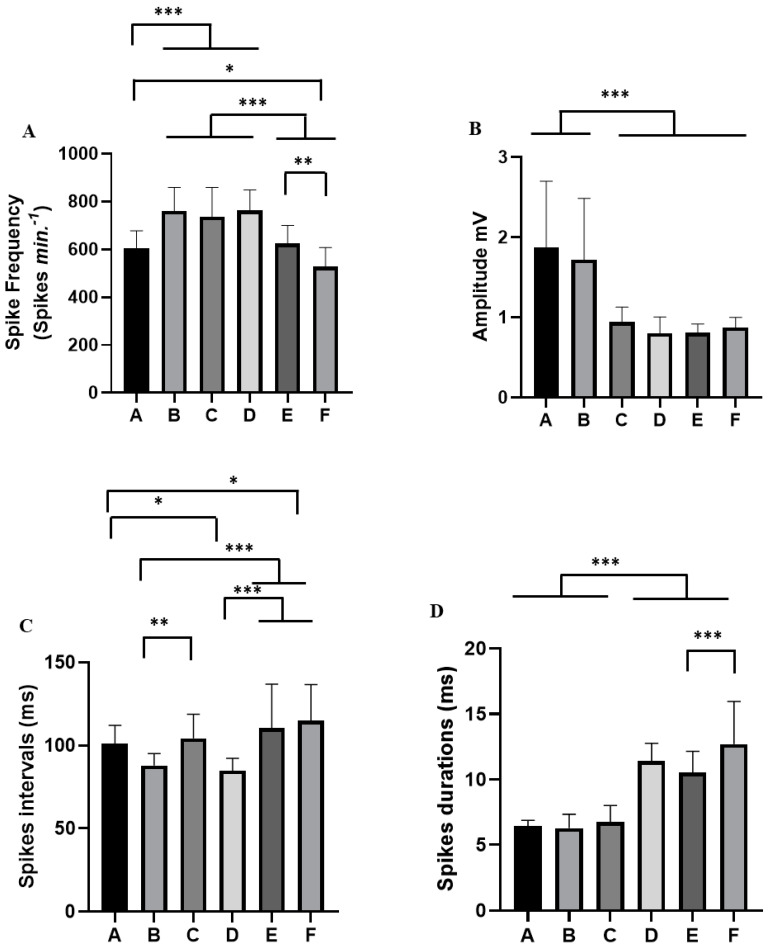
Evaluation of the cardiac activity of *Melipona flavolineata* during physical restraint in periods A, B, C, D, E and F: (**A**) peak frequency (Spikes min^−1^); (**B**) amplitude (mV); (**C**) peak intervals (ms); and (**D**) spike durations (ms) (after ANOVA followed by Tukey. * *p* < 0.05, ** *p* < 0.01, *** *p* < 0.001 (n = 8)).

## Data Availability

Data will be made available on request.

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
