# Peer review of "First Records of Heartbeats via ECG in a Stingless Bee, Melipona flavolineata (Apidae, Meliponini), during Contention Stress Using Isoflurane as an Anesthetic"

_insects, 2023, doi:10.3390/insects14080696_

Round 1

Reviewer 1 Report

The methodological part is incomplete and inconsistent:

MEASUREMENT:
Measurement technology is not described consistently. It is clearly described that electrodes were homemade, and placed by a stereotactic equipmemt. But, did the experimenter test the quality of the electrode(s)? Was always the same electrode used?

The amplifier device is named, and the amplification factor (50000), but it is not determined at which time resolution (Hz) the data were digitized, and which digitizing resolution was chosen (8,10,12,16 .. bit?

Which filters and settings have been used? Were the characteristic points of ECG determined manually or automatically by algorithms or commercial biosignal software? In such case, was there a manual overreading and correction process applied?.

STATISTICS

This section is incomplete and inconsistent.

Statistical measures (mean, SD and n) should be mentioned here. It is not acceptable that the reader has to search for the meaning by reading the whole article to find out.

Line 147 mentions that normality has been tested by  Kolmogorov-Smirnow test. But the report never mantions any results of this test. Homogeneity test by Levene is mentioned, too. Buit nowhere any results reported. 

Statistical methodology section mentions "two-way ANOVA, followed by Tukey's test", but the reader must guess the factors as they are not described. Clearly, one factor is the TIME segment (A-F). But the second factor - the bee? How about interactions?

Line 152 mentions that an unspecified procedure of GraphPadPrism (Version 8) is used to identify and remove outliers.

Methodologically seen, this is a very weak approach, specifically ad it relates to  a very low number of just 8 individuals, and is largely dependent on the distribution and homogeneity assumptions.  A much better approach would be  rely on robust statistical methods with transformation (like Box-Cox), or GLMs.

RESULTS

The number separators ("," for thousands and "." decimal points) are not always used correctly. Examples: line 189; line 169.

The number of significant decimals should not suggest inadequate precision. Examples: line172: "QRS duration averaged 6.15+-0.2268 ms" wOuld be better represented by "6.3+-0.23 ms"
Line 2020: "mean of 596+-81.91 bpm" would be better pesented by "596+-82 bpm"

ECG part

It is not discussed that the flow of haemolymph may chasnge direction. But, even if not measured, this should be mentioned. 

Rhythmicity of heartbeat: At least in human ECG, it is standard to represent rhythm changes by th Poincaré diagram [plot (R-1 - R) by (R - R+1) ]

GENERAL REMARK

Overall, an interesting paper.

For me, having analyzed several thousand Holter ECGs in my lifetime, it was interesting to see how similar the human and bee ECGs look, despite of the anatomical&species physiology differences.

The exploration of rhythm changes would be an  interesting project.

Btw, the recovery time after after a stop of isoflurane anesthesia in humans is about 3 - 6 minutes - similar to bees.

A few efforts to revise this paper might improve quality and readability of this paper.

Author Response

Dear Referee,

We answered all your concerns in the manuscript. Here are the details of our modifications.

Measurement technology is not described consistently. It is clearly described that electrodes were homemade, and placed by a stereotactic equipmemt. But, did the experimenter test the quality of the electrode(s)? Was always the same electrode used?

A: See answer in lines 128-129.

The amplifier device is named, and the amplification factor (50000), but it is not determined at which time resolution (Hz) the data were digitized, and which digitizing resolution was chosen (8,10,12,16 .. bit?

A: LINES 136-139 and Figure 2.

Which filters and settings have been used? Were the characteristic points of ECG determined manually or automatically by algorithms or commercial biosignal software? In such case, was there a manual overreading and correction process applied?.

A: LINES 153-156

This section is incomplete and inconsistent.

A: we do not agree with this statement, since all anaylsis were properly detailed.

Statistical measures (mean, SD and n) should be mentioned here. It is not acceptable that the reader has to search for the meaning by reading the whole article to find out.

A: The measures we analyzed were mentioned. LINES: 155-160

Line 147 mentions that normality has been tested by  Kolmogorov-Smirnow test. But the report never mantions any results of this test. Homogeneity test by Levene is mentioned, too. Buit nowhere any results reported.

 R : the Kolmogorov-Smirnov test was only used to test the null hypothesis for evaluating the groups, so it cannot be mentioned in the text.

Statistical methodology section mentions "two-way ANOVA, followed by Tukey's test", but the reader must guess the factors as they are not described. Clearly, one factor is the TIME segment (A-F). But the second factor - the bee? How about interactions?

R: Cardiac functioning is distributed over time, and the ECG analysis depends on the amplitude of the signal and the time in which it occurs.

Line 152 mentions that an unspecified procedure of GraphPadPrism (Version 8) is used to identify and remove outliers.

R: It was used for statistical analysis and evaluation of mean QRS Duration, QT, RR and frequency.

Methodologically seen, this is a very weak approach, specifically ad it relates to  a very low number of just 8 individuals, and is largely dependent on the distribution and homogeneity assumptions.  A much better approach would be  rely on robust statistical methods with transformation (like Box-Cox), or GLMs.

A: We do not agree, because the results between the animals were very similar with similar morphographic elements, which can be proven when observing the standard deviation between the means obtained. The sample number was appropriate for the taxon we studied.

The number separators ("," for thousands and "." decimal points) are not always used correctly. Examples: line 189; line 169.

A: Corrected

The number of significant decimals should not suggest inadequate precision. Examples: line172: "QRS duration averaged 6.15+-0.2268 ms" wOuld be better represented by "6.3+-0.23 ms"

A: Corrected in all points in which this issue appeared.

Line 2020: "mean of 596+-81.91 bpm" would be better pesented by "596+-82 bpm"

A: Corrected

It is not discussed that the flow of haemolymph may change direction. But, even if not measured, this should be mentioned.

A: We included a whole section in the discussion in which we appreoached this subject. LINES:  309-330

Rhythmicity of heartbeat: At least in human ECG, it is standard to represent rhythm changes by th Poincaré diagram [plot (R-1 - R) by (R - R+1) ]

A: We obtained the frequencies of the selected areas in the probe recording, where the mean and standard deviation of the heart rates obtained during the selected time intervals in the recording were obtained, which was adequate in studies of invertebrate circulatory patterns.

Reviewer 2 Report

Understanding the cardiac activity of bees is usefully method to evaluate the effect of  stressors on animals. This ms studied the hemodynamic activity of Melipona flavolineata under restraint stress and used isoflurane as an anesthetic. The results could provide important information for evaluation of toxic agents on the physiological issues of model species. This ms was prepared well, and it can be accepted after miner revision.

L22-23,deleted and

L24-25, The authors should provide the purpose of this study in the abstract.

Understanding the cardiac activity of bees is usefully method to evaluate the effect of  stressors on animals. This ms studied the hemodynamic activity of Melipona flavolineata under restraint stress and used isoflurane as an anesthetic. The results could provide important information for evaluation of toxic agents on the physiological issues of model species. This ms was prepared well, and it can be accepted after miner revision.

L22-23,deleted and

L24-25, The authors should provide the purpose of this study in the abstract.

Author Response

Dear reviewer,

Thanks for the comments. As requested, we included the purposes of the study (lines 13-14) and corrected lines 22-23 (it was lacking the word "cold").

We performed an edition of the language, as requested by you, the editor, and the other referee.

Sincerely

Author Response

  1. The signal is collected by two electrodes which the recording position is unclear. Are the electrodes simply placed on the surface of the abdomen or have holes been drilled to enable the electrodes to plunge into the hemolymph in very closed contact with the dorsal vessels or penetrate the heart tissues? Clear description is needed and more accurate references should be added in the introduction that enable readers to distinguish between these different approaches in insects (intracellular recordings, bipolar electrode, etc.)

A: We provided the requested information (the electrodes punched the abdomen, in the already described position 0.6 mm – Lines 134-136, Fig. 2). More information in ECG in insects are inappropriate since they are a huge diverse group, with variations in the hemolymph flow. The available information, for bees, is provided in the introduction and in the discussion.

  1. The analysis based on the properties of human (mammalian) ECG is inappropriate. In human ECG, multiples derivations, usually 10, is applied to extract positive and negative waves called P, Q, R, S and T. The physiology of the human heart in two parts with auricle and ventricle is very different from the insect heart with a tubular structure made of successive chambers. Consequently, the parallel with the human (mammalian) waves is not relevant and should be removed from the manuscript. Comparative physiology of insect and human hearts was undertaken by Karel Sláma.

A: The paper that best described the ECG was the one by  Walter Kaiser, Theo Weber, Dietmar Otto, Anton Miroschnikow. Oxygen supply of the heart and electrocardiogram potentials with reversed polarity in sleeping and resting honey bees. Apidologie, 2013, 45 (1), pp.73-87. 10.1007/s13592-013-0229-2 hal-01234706.  The mentioned paper showed (Fig. 4), for A. mellifera, a registry like the one we described. We also observed a likeliness on QRS during the contraction of the tubular structure (depolarizarion). Following the position of the electrodes (Fig 2), we captured the depolarization and the repolarization of the tubular structure, which is conventionally called T-wave. There is no registry of P-wave in our recordings.

  1. Beside these denominations, the “QRS” waves represent clearly the heartbeat of the Melipona depending the position of the electrodes. This signal is in accordance with others, particularly the work of Kaiser & al, 2013 (which is cited with the wrong year. It was published in 2013 not 2014). But I'm not convinced that the “T” wave represent the repolarization of the heart. It's a matter of over-interpretation that lacks the scientific evidence to be affirmed. The reviewer’s speculation is that correspond to the activity of others muscles in the abdomen (especially since the work is carried out with awake animals). This point is key because a part of the analysis is depending of the origin of the T wave. Consequently the authors should provide solid evidence indicating that the T-wave originates from the dorsal vessel.

A: The online version (just after acceptance) was published in 2013, but the final version was published in 2014, as shown in the website (https://link.springer.com/article/10.1007/s13592-013-0229-2). Figure 5 (A to F) shows that is a correlation between the appearing of the depolarization components followed by a repolarization wave. Since we used stereotaxic, fixed, coordinates, our data are comparable between the samples (bees), and eased the signal acquisition.

  1. PSD analysis is a good way to identify heartbeat modulations. However a section in M&M should be added to describe the PSD methods. In view of the results obtained in Fig. 4, we may also question about the relevance of choosing 6 seconds of analysis, systematically distributed throughout the recording. Five periods of cardiac activity are remarkable (0-200s, 200-600s, 600-800s, 800-1000s and 1000- to the end). Wouldn't it have been more appropriate to choose 2 or 3 seconds of analysis in each of these periods which could correspond to specific physiological states after anesthetic episodes (or contention stress)? It would be worthwhile to try these analyses.

A:  We do not agree with these suggested periods since we have shown that 1 second in the periods we tested (299-300s, B= 599-600s, C=899-900s, D= 1199-1200s, E= 1499-1500s and F=1799 -1800s) was enough to show significant patterns. Our registers lasted 1,800 seconds, and showed different answers related with the restraining stress (frequency, amplitude and intervals); PSD was used to analyze the power of each 5 minute period of cardiac stress. Figure 4 shows the consequencies on the hemodynamic of the animal.

  1. The claim that the regulation of cardiac activity described here is an adaptation to a state of stress due to restraint is speculative and lacks evidences. The change in cardiac activity profile is probably due to recovery from anesthesia. These claims should be removed from the paper.

A: Our study was pioneer in this subject in this group of eusocial bees; thus, future studies must deepen the discussion on the effects of isoflurane and stress, separately, and combined. The concerns raised by the referee were included in the manuscript (lines 288-303). Since these information’s are new, we do not agree to remove this point, but we better developed its interpretation.

  1. Similarly, the conclusion that isoflurane is an anesthetic with fewer effects on melipona physiology is clearly not demonstrated in the work presented. A comparison with other types of anesthesia is required.

A: We included this subject in the discussion (lined 307-316), and as already pointed, the present study is novel, and future studies will specifically explore it.

Round 2

Author Response

Dear Reviewer,

We addressed all the questions you raised. 

Point 2&3: As I already indicated, I do believe that the QRS complex recorded correspond to the heart beat and the signal is in accordance with what have been already record on honeybee or others insects. However, while ECGs in melipods are a novelty, ECGs in insects are a technique that has already been
used in several models. To my knowledge this repolarization phase has never been described before. If the authors want to claim that the wave T is a macroscopic repolarization they have to demonstrate it. For example, Papaefthimiou & Theophilidis (Journal of Insect Physiology 57 (2011)
316 325) have used cardiac-CAP to describe a repolarization phase. On the other hand, it has been shown that other muscular contractions (sometimes rhythmic) can be surimposed to cardiac electrical activity. The figure 5 of Kaiser&al (Walter Kaiser, Theo Weber, Dietmar Otto, Anton Miroschnikow. Oxygen supply of the heart and electrocardiogram potentials with reversed polarity in sleeping and
resting honey bees. Apidologie, 2013, 45 (1), pp.73-87. 10.1007/s13592-013-0229-2 hal-01234706) show clearly that potentials from others « unidentified muscles » could appears in insect ECG recording. It's highly likely that this noisy 'T' wave is the result of a muscle contraction of some kind, and has nothing to do with repolarization. Clear evidences correlating repolarization phase and T
wave of ECG must be provided. Finally I still think that using the PQRST terminology of mammalian ECG is misguided and leads to confusion.

A: We removed the T-wave from our results and from the other parts of the manuscript. We agree that the pattern we found must be futher explored and investigated. We also modified the text and figures regarding the PQRST terminology in mammals, in order to make it clear that insects have a different pattern. We modified all the results in order to incorporate these changes. 

Point 4:
PSD methods are still missing from the M&M section.

Just because the authors found "significant patterns" doesn't mean they found the physiological relevance of their study. Work on more than these 6 seconds of analysis should be carried out. For each period, they should produce replicates (at least 3 seconds per period) to demonstrate the relevance and homogeneity of their results, which will then enable us to understand cardiac adaptations to their experimental conditions.

A: We made these replicates, as requested, and showed that, in the periods we delimited, the results are similar.

Point 5 & 6:
I understand from the authors' response that they are admitting the absence of evidences to assert that they observed cardiac adaptation to restraint stress and that isofurane is an anesthetic with fewer effects on melipona physiology. In this case, it is imperative to remove from the manuscript all (hasty) conclusions concerning the adaptation of the heart to stress and the preferential use of isofurane for Melipona anesthesia.

A: We removed these conclusions, and made it clear that the use of isoflurane in the stingless bee (specially Melipona) must be further investigated.

Round 3

Reviewer 3 Report

It's disappointing that, instead of limiting ourselves to textual adaptations/changes, this could have achieved even more convincing results by adopting a few analyses or manips.